# Application of Electrochemical Oxidation for Water and Wastewater Treatment: An Overview

**DOI:** 10.3390/molecules28104208

**Published:** 2023-05-20

**Authors:** Mohammad Saleh Najafinejad, Simeone Chianese, Angelo Fenti, Pasquale Iovino, Dino Musmarra

**Affiliations:** 1Department of Engineering, University of Campania “Luigi Vanvitelli”, Via Roma 29, 81031 Aversa, Italy; mohammadsaleh.najafinejad@unicampania.it (M.S.N.); simeone.chianese@unicampania.it (S.C.); 2Department of Environmental, Biological and Pharmaceutical Sciences and Technologies, University of Campania “Luigi Vanvitelli”, Via Vivaldi 43, 81100 Caserta, Italy; pasquale.iovino@unicampania.it

**Keywords:** emerging pollutants, wastewater treatment, electrochemical oxidation, role of operational parameters, cost analysis assessment, photoelectrocatalysis

## Abstract

In recent years, the discharge of various emerging pollutants, chemicals, and dyes in water and wastewater has represented one of the prominent human problems. Since water pollution is directly related to human health, highly resistant and emerging compounds in aquatic environments will pose many potential risks to the health of all living beings. Therefore, water pollution is a very acute problem that has constantly increased in recent years with the expansion of various industries. Consequently, choosing efficient and innovative wastewater treatment methods to remove contaminants is crucial. Among advanced oxidation processes, electrochemical oxidation (EO) is the most common and effective method for removing persistent pollutants from municipal and industrial wastewater. However, despite the great progress in using EO to treat real wastewater, there are still many gaps. This is due to the lack of comprehensive information on the operating parameters which affect the process and its operating costs. In this paper, among various scientific articles, the impact of operational parameters on the EO performances, a comparison between different electrochemical reactor configurations, and a report on general mechanisms of electrochemical oxidation of organic pollutants have been reported. Moreover, an evaluation of cost analysis and energy consumption requirements have also been discussed. Finally, the combination process between EO and photocatalysis (PC), called photoelectrocatalysis (PEC), has been discussed and reviewed briefly. This article shows that there is a direct relationship between important operating parameters with the amount of costs and the final removal efficiency of emerging pollutants. Optimal operating conditions can be achieved by paying special attention to reactor design, which can lead to higher efficiency and more efficient treatment. The rapid development of EO for removing emerging pollutants from impacted water and its combination with other green methods can result in more efficient approaches to face the pressing water pollution challenge. PEC proved to be a promising pollutants degradation technology, in which renewable energy sources can be adopted as a primer to perform an environmentally friendly water treatment.

## 1. Introduction

Water has a crucial role in the life of all living beings and their survival; therefore, attention to water quality should always be considered. Over recent years, the presence of emerging and persistent pollutants, such as microplastics, antibiotics and pharmaceutical compounds, pesticides and ammonia [1,2,3,4,5,6], is increased in liquid effluents and groundwater. Physical, chemical, and biological methods can be used to remove these pollutants, but the right treatment choice is very prominent. In recent years, many methods have been employed to treat different wastewater, such as decolorization of colored wastewater using green nanocomposites, floating treatment wetland (FTW), adsorption, membrane filtration, and solvent extraction [7,8,9,10,11], among others. However, each method has advantages, disadvantages, and limitations regarding efficiency, cost, feasibility, and environmental impact. Although there is currently no reliable and direct answer to the question of which is best, over the past few years, very effective methods have been used to treat wastewater, such as the so-called advanced oxidation processes (AOPs), because of their efficient, cost-effective, environmentally friendly character, and high capability to remove a wide range of pollutants [12]. AOPs include different techniques, such as ozonation, sonolysis, photocatalysis, fenton and photo-fenton, and electrochemical oxidation [13]. The common denominator of AOPs is the generation of hydroxyl radicals (**^•^**OH), which are very effective in the degradation of highly toxic and persistent pollutants, as they are among the strongest oxidants, with a very high standard oxidation potential E (**^•^**OH/H_2_O) = 2.80 V_SHE_ [14]. Therefore, the efficiency of advanced oxidation systems is strictly affected by the amount of **^•^**OH generated along the process. Besides **^•^**OH, other reactive oxygen species (ROS) can be generated, which can degrade pollutants, including HO_2_**^•^**, ^1^O_2_, O_2_**^•^**^−^, and SO_4_**^•^**^−^ [15], as shown in Figure 1.

In recent years, much attention has been paid to a specific and efficient AOPs technology called electrochemical oxidation (EO), which is widely employed to remove hazardous, recalcitrant, and emerging contaminants [16]. Hence, this review provides an overview of EO and describes a detailed analysis of various aspects of this process. First, the main operating parameters in EO were investigated, including anode materials, current density, composition, and electrolyte concentration. Some parameters that were less considered in previous studies, such as temperature, initial concentration of wastewater, and distance between electrodes, were examined to define the optimal conditions for completely removing contaminants. Moreover, the batch and continuous reactor configurations, the kinetics of electro-oxidation reactions, and cost analysis were discussed. Finally, the review aims to provide a brief introduction regarding the combination method between EO and photocatalysis (PC), called photoelectrocatalysis (PEC). This process proved to be a promising technology, but more studies are required to scale up to industrial-scale applications.

### Methodology and Search Strategy

The keywords “electrochemical oxidation”, “emerging pollutants”, “electrochemical oxidation and impact of operational parameters”, “electrochemical oxidation and cost analysis”, “electrochemical advanced oxidation processes”, and “photoelectrocatalysis” were introduced in various databases, including Scopus, Google Scholar and SpringerLink, to identify a list of peer-reviewed reviews and scientific articles related to removing emerging pollutants from impacted water by means of EO and PEC. The research was carried out mainly by collecting studies from the last 10 years. Anyway, milestone EO studies were referenced due to their undeniable scientific relevance.

## 2. Electrochemical Oxidation (EO)

Electrochemical oxidation (EO) has recently gained much attention as a very effective method for removing a wide range of pollutants. Efficiency and green performances are the main reasons this technique is very reliable and widely used [17]. EO processes are characterized by two approaches called direct electro-oxidation, or anodic oxidation, and indirect electro-oxidation, as discussed in Section 3. Briefly, in the direct EO process, a direct electron transfer (DET) between the anode surface and contaminants results in pollutants degradation [18]. On the other hand, indirect EO involves the homogeneous reaction of organic pollutants with strong oxidants, such as Cl_2_, H_2_O_2_, HClO, ClO^−^, SO_4_^2−^, and O_3_, among others, produced during the electrolysis process.

### 2.1. Study of Electro-Oxidation Operating Parameters

The performances of the EO process are affected by several operational parameters, including initial contaminant concentration, anode materials, supporting electrolyte, electrolyte concentration, current density, and temperature, among others.

#### 2.1.1. Effect of the Anode Material

In recent years, many EO studies have been carried out by means of a variety of anode materials. In the EO processes, since the amount of **^•^**OH produced is dependent on the nature of the anode material, the choice of anode material with high electrocatalytic activity is very important to obtain high oxidation rates [19]. The selected anodic materials must have high oxidation evolution potential and be highly active against organic oxidation, stability in an electrolysis environment, long life, cost-effectiveness, and environmental friendliness. Anodic materials are generally classified into two classes, active or non-active anodes, based on the capability to bond **^•^**OH on their surface. Active anodes establish a strong interaction with **^•^**OH, which result only in partial degradation of the target contaminant. Conversely, non-active anodes weakly bond **^•^**OH, leading to complete substrate mineralization [20]. Figure 1a shows the standard classifications and properties of typical active and non-active anodes. As shown, the non-active anodes can accumulate more hydroxyl radicals on their surface, making them more efficient in EO treatment of impacted water. Particular non-active anodes, including BBD, PbO_2_, Ti_n_O_2n−1_, and SnO_2_, own a high oxidation power [20]. Several anodes were employed in different EO studies in recent years, and BDD was the most used material (Figure 1b). The advantages and disadvantages of several common anodes are briefly collected in Table 1.

#### 2.1.2. Effect of the Supporting Electrolyte and Electrolyte Concentration

As discussed above, the performances of the EO process are influenced by both the supporting electrolyte type used and the initial electrolyte concentration. Various electrolytes are usually employed in EO treatments, such as sodium chloride (NaCl), sodium sulfate (Na_2_SO_4_), and sodium nitrate (NaNO_3_), among others. Since, in natural conditions, effluents have low conductivity, the need for a supporting electrolyte is important to increase the conductivity of the system. Therefore, attention to this item plays a very prominent role. Adding an electrolyte may increase the efficiency of EO and the conductivity of the wastewater solution. When specific supporting electrolytes are added to the solution, strong mediator oxidants can be produced that can oxidize pollutants by indirect EO [6]. The different behaviour of electrolytes in the EO process can be attributed to their different performance with the produced oxidizing species due to their different oxidation power [27]. By adding an electrolyte, the electric current flowing through the cell increases because the electrical conductivity increases and the resistance decreases; thus, power consumption can be reduced, which will positively affect the wastewater treatment system (Equation (1)) [28]: P = R × I^2^ = I × V(1)
where R is the resistance (Ohm), V is the voltage (Volt), P is the power consumption (Watt), and I is the electric current (Ampere).

Table 2 shows several works carried out in recent years focused on PFASs (PFOS and PFOA) removal from impacted waters to determine what supporting electrolyte in each article was mostly used.

In several previous EO studies, different electrolytes (Na_2_SO_4_, NaBr, NaNO_3_, and NaCl) were tested, keeping the other operating parameters constant, to evaluate the electrolyte composition’s effect [34]. As shown in Figure 2a, many studies reported that Na_2_SO_4_ is the more efficient supporting electrolyte for COD removal. Two different concentrations (5 and 7 g/L) were selected for Na_2_SO_4_, which showed that by increasing the concentration from 5 g/L to 7 g/L, the contamination removal increased from 87% to 91% [34]. The effects of NaCl and Na_2_SO_4_ were investigated under the same initial concentration of COD, phenols, cyanide, and ammonia, and it was observed that NaCl had a better performance than Na_2_SO_4_. The percentage of contamination removal for electrolytes NaCl and Na_2_SO_4_ was 90% and 60%, respectively (Figure 2b). By increasing NaCl concentration, the pollutant removal increased, but from the range of 1.6 to 2.4, this value was almost fixed, highlighting that the best NaCl concentration of 1.6 g/L was identified as the optimum concentration (Figure 2c) [35]. Another study used different electrolytes to remove p-nitrophenol (PNP) using EO by Ti/Ti_4_O_7_ anode. Figure 2d shows the percentage of pollution removal operating in the presence of NaCl, Na_2_SO_4_, NaClO_4_, Na_3_PO_4_, and NaNO_3_, respectively [36].

#### 2.1.3. Effect of the Current Density

Applied current density is another of the main parameters affecting EO processes, because it plays a drastic role in controlling the generation rate of the oxidizing species [37]. Current density is a vector quantity defined as the amount of electric current flowing through a unit cross-sectional area [35]. Current density can be expressed by the following formula (Equation (2)):i = I/A(2)
where i is the electric current density (mA/cm^2^), I is the electric current flowing through a given material or a conductor (A), and A is the cross-sectional area of a material or a conductor (m^2^).

In general, the contaminant degradation efficiency increases in the EO process by increasing the applied current density. It should be noted that an increase in current density increases the cost of EO due to the increase in energy consumption, so finding the optimal point could result in an energy-saving strategy. However, an increase in current density does not always enhance the oxidation rate for increasing the removal efficiency of pollutants due to the occurrence of the oxygen evolution reaction and/or mass transport limitations [38,39]. Therefore, many works have investigated the changes in current density on the efficiency of the EO reaction and finally suggested the optimal current intensity, shown in Table 3.

In a previous study by Song et al. [45], different current densities (5, 10, and 15 mA/cm^2^) were applied to degrade Sulfamethazine at initial COD = 50 mg/L; it was observed that the removal rate of pollutants increased with increasing current density, and the removal efficiency increased from 68.40% to 94.28% and 99.25%, respectively. In another study by Sanni et al. [34], different current densities of 50, 80, 120, 150, and 230 mA/cm^2^ were applied, using BDD as anode and graphite as a cathode in an EO process to remove various contaminants, over a treatment time of 4 h. The highest removal efficiency achieved was about 96% operating at a current density of 230 mA/cm^2^. Anyway, the optimum current density identified was 120 mA/cm^2^, because, acting under these conditions, both excellent removal efficiency and low energy consumption were achieved. On the other hand, power consumption increased from 85 kWh/m^3^ to 785 kWh/m^3^ as the current density increased from 50 A/cm^2^ to 230 A/cm^2^. Figure 3 shows four different current densities of 4, 6, 8, and 10 mA/cm^2^ that were tested with Ti/Ti_4_O_7_ as the anode. Increasing the current density from 4 to 6 mA/cm^2^ significantly increased both p-nitrophenol removal percentage and PNP removal kinetics from 0.0762 to 0.548 min^−1^. Then, with increasing the current density from 6 to 10 mA/cm^2^, there were no drastic changes; thus, to prevent further energy consumption, the current density of 6 mA/cm^2^ was recognized as the optimal point [36].

#### 2.1.4. Effect of the Other Operating Parameters

Although other parameters such as reaction temperature, initial concentration of contaminants, and the effect of distance between electrodes have been less studied, they can certainly affect the removal efficiency of contaminants in EO technology [46]. The temperature can affect the removal efficiency of pollutants, but direct oxidation processes have not much impacted by it. In general, when temperature increases, the efficiency of removing pollutants should increase since it causes a greater mass transport toward the anode because of the decrease of medium viscosity [47,48]. Flox et al. [49] studied the effect of temperature in an EO process to remove the herbicide mecoprop from water. The results showed that by increasing the temperature from 15 °C to 60 °C, the charge rate decreased from 35 A h dm^−3^ to 20 A h dm^−3^, and the efficiency and removal rate of TOC were performed faster. Conversely, in a previous study [50], an enhancement in temperature resulted in a decrease in the contaminant degradation performances (Figure 4a). In this study, BDD was used as an anode to remove COD, and the current density was equal to 20 mA/cm^2^. EO operations at ambient temperature are generally preferred [51]. The type of water and the initial concentration of wastewater are particularly important in the EO process. In general, water supplies have different organic and mineral substance loads, which affect the percentage of pollution removal. Most experimental studies have focused on ultrapure and pure water [37,52], but these unrealistic systems completely differ from the real conditions. In a Panizza et al. [48] study, the effect of initial COD concentration (0.1–0.3 mM) was investigated, acting with BDD anode at a constant current density of 40 mA/cm^2^, T = 25 °C and a flow rate of 300 dm^3^/h. In this study, the current efficiency percentage increased by increasing the initial COD concentration. Zhuo et al. [53] examined the effect of the initial perfluorooctanoic sulfonate (PFOS) concentration on the removal efficiency of the EO system. The findings (Figure 4b) displayed that when PFOS concentration increased from 100 mg/L to 300 mg/L, the removal ratios at the anode increased from 61.87% to 82.96%, likely due to the higher chances of PFOS reaching the anode surface. In another study [54], the effects of the initial concentration of PFOA were determined, operating with Ti/SnO_2_-Sb/Yb-PbO_2_ anode at four different initial PFOA concentrations of 10, 50, 100, and 200 mg/L. The results showed that with increasing initial PFOA concentration, the electrochemical reaction rates increased.

Another important factor influencing the EO process is the distance between the electrodes [55]. By reducing this distance, the removal efficiency of organic matter usually increases. This is because it is one of the important factors in reducing cell resistance, which decreases with decreasing distance between electrodes. In other words, reducing the distance between the electrodes requires less electrical energy to transfer ions, because the cell resistance is low [56,57]. In a previous study [58], five different electrode gap distances (0.003, 0.005, 0.007, 0.009, and 0.011 m) were considered to define the optimal conditions. Keeping the other operating conditions constant, the pollutant removal efficiencies obtained have been 91, 85, 79, 69, and 74%, respectively. As a result, the optimal distance between the electrodes was identified as 0.003 m due to lower energy consumption and higher removal efficiency achieved. In another study by Ma et al. [54], the effect of electrode distance was measured to remove PFOA, and finally, the optimal distance between the electrodes was obtained. The electrode distances range investigated was among 5, 10, 15, and 20 mm, and the degradation ratios of PFOA were 95.11 ± 3.9%, 89.05 ± 1.1%, 85.23 ± 1.4%, and 67.92 ± 3.8%, respectively. As shown in Figure 4c, the optimal distance of 5 mm was selected.

### 2.2. Electrochemical Reactor Design

The design of an electrochemical reactor is one of the significant factors affecting the performance of EO processes for removing emerging pollutants from wastewater [59]. Depending on the type of operation mode, electrochemical reactors can be divided into batch- or continuous-flow reactors. However, several parameters, such as ohmic drop, generation of gas bubbles on the electrodes, and mass transfer, must be carefully considered while designing an EO reactor. Among others, the batch operation mode, using 2D vertical parallel electrode configuration, is the reactor design more employed in EO processes due to its easiness of installation, handling, and sampling. Several studies report high organic EO degradation efficiencies operating with this design [17,60]. Magro et al. [61] studied the EO process to treat triclosan and its degradation by-products using BDD vertical parallel electrodes in a batch reactor, achieving around 100% degradation efficiency after 240 min. Hao et al. [62] used a conventional single-compartment reactor with vertical parallel plates for degrading phenol from synthetic water. The results indicated 86% of TOC removal after 60 min. The same reactor configuration has been used by Periyasami et al. [63] for the EO removal of florfenicol antibiotic from synthetic water, obtaining 91% of TOC removal after 180 min. Calzadilla et al. [64] studied the anodic oxidation of 30 pharmaceuticals, achieving 90% of mineralization efficiency after 330 min. Although the application of batch mode enables excellent EO performances for degrading organics, several downsides, such as mass transfer limitation, control potential instability, poor current distributions, and complex hydrodynamics, remain. This configuration may lead to massive electrode passivation due to the formation of a gas-bubbling layer, resulting in adverse blockage of the electrode surface [65,66]. Some of these drawbacks may be overcome by employing batch-divided reactors, for example, by means of a membrane to separate the anodic and cathodic compartments [67,68]. This configuration is highly viable and enhances the current efficiency [69], but the space between the electrodes is very large, causing slower reactions [70]. Anyway, an undivided batch reactor, where both cathode and anode are in the same vessel, is still the most studied configuration, with its simple design and ease of scale-up to an industrial scale [71]. To summarize, the batch operation mode is a promising reactor configuration to be adopted for EO organic pollutants removal from water. However, some limitations are associated with slow mass and charge transfer, small wastewater volume to treat (lower than 1 L), and low potential for large-scale application [72].

For this proposal, continuous-flow reactors are better for treating large volumes of polluted wastewater and operate under more realistic conditions. Continuous-flow reactors enhance faradaic efficiency, causing lower energy consumption and lower electrolyte loads [70,73]. Moreover, using continuous-flow reactors on the laboratory scale permits an easier process scaling up [74]. In general, continuous-flow reactors can be divided into two main configurations, namely, flow-by and flow-through modes [75]. In the flow-by configuration, the electrodes are positioned in parallel to the flow direction, while in the flow-through, they are placed horizontally [76]. Operating EO in the flow-by or flow-through mode significantly impacts the mass transfer of several organic compounds [18]. However, several previous studies have reported that electrochemical flow-through reactors exhibited higher removal efficiencies of contaminants and lower energy consumptions than flow-by reactors due to the higher mass transfer efficiency; this configuration improved bulk solution reactions by means of reactive oxygen species [77,78]. Zicheng et al. [79] compared the two configurations for ammonium and COD removal from impacted marine aquaculture. Again, the flow-through configuration showed the best performance, showing a higher formation rate of free chlorine and, thus, the removal rate of contaminants. Figure 2 shows some of the main reactor configurations used in previous EO investigations. 

## 3. Mechanism of Electrochemical Oxidation of Organic Pollutants

As introduced above, the EO process may occur by two mechanisms: direct EO, where organics pollutants are oxidized by transferring electrons after adsorption on the anode surface, without the implication of any chemical substances; indirect EO, where strong oxidants (i.e., chlorine, hypochlorite, persulphate ion, or hydrogen peroxide) are formed on the anode surface and propagated in the bulk solution, resulting in the organic degradation of molecules [12].

### 3.1. Direct EO

Direct EO mechanism only involves the contribution of electrons, which oxidize the contaminants molecules, resulting in direct charge-transfer reactions [19]. The first step is the adsorption of the pollutants onto the anode surface. The process is mainly controlled by the molecules transport and electron transfer rate at the anode/solution interface [80]:R + M → M-R_ads_   (adsorption reaction)(3a)
M-R_ads_ + e^−^ → M-R_ads,ox_       (direct charge electron reaction)(3b)
M-R_ads,ox_ → R_ox_    (desorption reaction)(3c)
where R is the pollutant molecule, and M is referred to as the anode surface.

The mechanism is facilitated by acting at low applied anode potential to avoid the oxygen evolution reaction (OER) that occurs at 1.23 V_SHE_. However, under conditions of low and fixed anode potential, it is likely to encounter the poisoning effect, resulting in the formation of a polymer layer on the anode surface and decreasing the electro-catalytic activity. The poisoning effect depends on the nature of the anode surface and both concentration and properties of the organic pollutants. Performing EO in the potential region of water discharge (anode potential > 2.3 V_SHE_) could be a manner to avoid anode fouling. OTR at high anodic potential involves a first step in which the production of adsorbed hydroxyl radicals from water discharge occurs:MO_x_ + H_2_O → MO_x_[**^•^**OH] + H^+^ + e^−^(4)

MO_x_ represents general oxide anode surface sites for the adsorption of **^•^**OH. The adsorbed [**^•^**OH] (also called physisorbed “active oxygen”) may interact with the oxygen already present in the oxide anode, resulting in a possible transfer of oxygen from [**^•^**OH] to the lattice of the anode and generating the so-called higher oxide MO_x+1_ (also called chemisorbed “active oxygen”) [20]: MO_x_[^•^OH ] → MO_x+1_ + H^+^ + e^−^
(5)

The reaction reported in Equation (5) is strongly affected by anode materials [12]. Several anodes encourage partial and selective oxidation of pollutants (conversion), while others favour complete oxidation to CO_2_ (mineralization). Therefore, as discussed earlier, it is possible to discern between two limiting classes of electrodes, defined as “active” and “non-active” anodes [19]. Active anodes, with low oxygen evolution potential (OEP), enable the generation of chemisorbed active oxygen MO_x+1_, favouring the conversion or selective oxidation of organics: MO_x+1_ + R → MO_x_ + RO (6)

Conversely, non-active anodes with high OEP avoid the reaction in Equation (5) and favour the mineralization to CO_2_ [20]: MO_x_[^•^OH] + R → MO_x_ + CO_2_ + H_2_O + H^+^ + e^−^(7)

Several active and non-active anodes are anew listed in Table 4.

### 3.2. Indirect EO

Indirect EO leverages the intermediation of some strong oxidant reagents, electrochemically generated on the anode surface, which act as electron carriers between the anode and the organic contaminant in the bulk phase. While direct EO has some advantages, such as no need for additional chemicals, indirect EO is more efficient in preventing electrode passivation and enhancing the electro-catalytic activity. The generation of oxidizing intermediate species accounts for a pivot role for indirect EO. To optimize this step, the potential at which the intermediates are generated must be quite far from the OEP; moreover, the rate for the intermediate generation must be higher than the rate of other side reactions, and then the contaminants molecules in the bulk phase must not be adsorbed on the anode surface, interfering with the direct EO pathway [81]. Typical oxidizing species are hydrogen peroxide, peroxydisulfate and sulfate radicals, ozone, and percarbonate [82]. Among others, active chlorine is the most traditional anodically oxidizing species employed for indirect EO of pollutants. Naturally occurring or added chloride available in solution may trigger some reactions to form electroactive chlorine species, such as chlorine gas, hypochlorous acid, or hypochlorite ion [83]: 2Cl^−^ → Cl_2(g)_ + 2e^−^(8)
Cl_2(g)_ + H_2_O → HOCl + Cl^−^ + H^+^(9)
HOCl → OCl^−^ + H^+^(10)

The rate of the reaction shown in Equation (9) is generally lower operating in acidic media due to OH^−^ instability and significantly higher in basic solution due to the fast formation of OCl^−^ (pK_a_ 7.44) ion, as reported in Equation (10), implying that the basic or neutral pH conditions are more desirable for degrading organic pollutants involving chlorine. When OCl^−^ is generated in the bulk phase, it takes action as the main oxidizing chlorine-mediator agent in the degradation of pollutants: Pollutants + OCl^−^ → CO_2_ + H_2_O + Cl^−^ + Products(11)

Equations (8)–(11) provide a cycle in which different electroactive chlorine species are formed [84]. These species were used to remove a wide range of organic contaminants from impacted water, showing high degradation efficiencies. In a recent study, Salvestrini et al. [17] worked on EO of humic acids operating with NaCl (10 mM), among others salt. The results showed that the presence of Cl^−^ allows the achievement of the highest humic acid degradation efficiency. The experiments were conducted at pH 10, entailing that ClO^−^ dominated over the other species and was probably the main driver of pollutant degradation.

In general, it has been shown that operating at 25 °C, Cl_2_ hydrolysis (Equation (9)) is complete at a pH higher than 4, and HOCl and OCl^−^ are the main available active chlorine species in solution in the pH range between 6 and 9. Hypochlorous acid and hypochlorite ions own a high oxidizing power due to the ClO^−^ bond polarization. Consequently, the reactions of HClO/ClO with organics may be depicted as oxidation, addition, or electrophilic reactions [85]. Anyway, albeit the use of active chlorine species in indirect EO processes enhances the general performances of the treatment, several problems exist, such as the possible formation of chlorinated organic intermediates or the low amount of chloride in the real water bodies. This involves that indirect EO needs the addition of salt to increase the process efficiency.

As reported above, peroxydisulfate (S_2_O_8_^2−^) and sulfate radicals (SO_4_^2−^) are also employed in indirect EO processes to degrade organics. Unlike chlorides, sulfates are not considered to be pollutants; hence, their use is desirable. S_2_O_8_^2−^ can be electrochemically generated from the oxidation of SO_4_^2−^ at an applied potential higher than 2 V_SHE_: 2HSO_4_^−^→ S_2_O_8_^2−^ + 2H^+^ + 2e^−^
(12)
2SO_4_^2−^ → S_2_O_8_^2−^ + 2e^−^(13)

As is well known, the reactions of S_2_O_8_^2−^ with organics at room temperature are not effective; thus, peroxydisulfate must be activated. Several methods are employed to activate S_2_O_8_^2−^, such as metal reactions or energy-based processes (heat, UV). Sulfate radicals have unique features, such as the ability to be a strong electron acceptor, enabling them to degrade persistent organic compounds successfully. Cai et al. [86] have applied an indirect electrochemical oxidation process to degrade the 2,4-dichlorophenoxyacetic acid from water using electrochemically generated persulfate. The results showed the feasibility of the approach. Tian et al. [87] successfully removed phenol from impacted water by persulfate-mediated electrooxidation. Yet, in a recent effort, Song et al. [88] scrutinized the electrochemical oxidation of organics in sulfate solutions. The presence of peroxydisulfate enhanced the process efficiency. The degradation rates of the organics were approximately doubled in the electrochemical activation of the peroxydisulfate process with respect to the experiments conducted in the absence of S_2_O_8_^−^.

To summarize, indirect EO implementation for degrading organic pollutants in impacted water may overcome several problems widely encountered in direct oxidation, such as anode surface fouling, mass transfer limitations, and higher energetic costs. However, specific studies are needed to seek the most suitable operating conditions, thus limiting several concerns such as the formation of undesired toxic by-products during the treatment [85].

### 3.3. EO Degradation Kinetics

The kinetics of the reactions provides information on the mechanism and the efficiency of the degradation process [89]. The order of the kinetics and the degradation rates depend on the configuration system and the operative parameters applied over the treatment. Several factors, such as the initial concentration of organic compounds, the concentration of the electrolyte, the applied current density, the pH of the solution, and the type of anode material, among others, may impact the degradation kinetics of pollutants [90]. These parameters strongly affect the generation of hydroxyl radicals, the mass transfer rate of pollutants, the availability of active sites, and, subsequently, the overall oxidation efficiency of the process [19]. By way of illustration, the degradation rate usually decreases with increasing the concentration of pollutants, because a greater quantity of organic molecules is forced to react with the same amount of oxidant species, resulting in an overall reduction of the oxidation rate [91]. Yet, the degradation rate typically increases significantly with increasing in the applied current density [92].

Generally, two different degradation kinetic models are adopted to describe the EO of organic pollutants in water: pseudo-zero-order kinetic (Equation (14)) and pseudo-first-order kinetic (Equation (15)). With the pseudo-zero-order kinetic, the process is considered under current control since the electron transfer process is hampered on the anode surface, resulting in current limitation conditions. Therefore, the rate of oxidation at the anode results slower than the rate of arrival of contaminant molecules: (14)$$\frac{d[C]}{d[t]}=-k_{0,app}$$
where *C* and $k_{0,app}$ are the contaminant concentration and the pseudo-zero-order kinetic constant, respectively. With the pseudo-first-order kinetic, the process is under mass transfer control since the rate of the oxidation is limited by the rate of diffusion of organic molecules to the anode surface (high current and low contaminant concentration) [93,94]:(15)$$\frac{d[C]}{dt}=-{k^{\prime }}_{obs}{\left[C\right]}^{m}{\left[OH\right]}^{n}$$
where *C* and **^•^**OH are the contaminant and **^•^**OH concentrations, *m* and *n* are the orders of the reaction with respect to the concentrations of *C* and **^•^**OH, respectively, and *k*′*_obs_* represents the apparent kinetic rate constant. As is well known, pseudo-first-order model assumes a quasi-stationary state for **^•^**OH radicals on the anode surface since they are very reactive and unable to be accumulated in the aqueous phase. This entails that a constant concentration of **^•^**OH always reacts with the contaminant (the reactant is considered in excess) on the anode surface during the full treatment time [95,96]. Based on the above considerations, Equation (15) can be rewritten as:(16)$$-\frac{d[C]}{dt}={k^{\prime }}_{app}{\left[C\right]}^{m}$$
where *k*′*_app_* = ${k^{\prime }}_{obs}{\left[OH^{∙}\right]}^{n}$, and *k*′*_app_* is the pseudo-first-order kinetic constant [97].

Most studies reported that the EO of organic pollutants exhibits pseudo-first-order kinetics, thus with the rate limited by mass transfer of the substrate from the bulk solution to the anode surface. Hai et al. [98] have studied the EO of sulfamethoxazole (SMX) using BDD anode. The results showed that the degradation of SMX followed pseudo-first-order reaction kinetics, and the reaction rates enhanced with the rising of the applied current density, with a maximum of 0.062 min^−1^ at 45 mA/cm^2^. Olver-Vargas et al. [99] investigated the EO of the dye azure B in an aqueous solution using BDD anode. The degradation of azure B obeyed pseudo-first-order reaction kinetics (*k_app_* of 0.37 min^−1^ after 500 min of treatment), suggesting that a constant amount of **^•^**OH reacts with the substrate at a given applied current density. Yet, Otzturk et al. [97] examined the EO of paracetamol (PCT) using Pt anode. Even in this study, the PCT degradation rate was well described by pseudo-first-order kinetic. The value of *k_obs_* increased linearly with increasing the applied current intensity from 0.02 to 2 A, proving a first-order dependence of the reaction on **^•^**OH. Brillas et al. [100] reported a study on the electrochemical incineration of Diclofenac (DCF) in aqueous media using Pt and BDD anodes. In both cases, the DCF decay followed a pseudo-first-order kinetic reaction, and the degradation rate was influenced by the type of anode material used during the treatment. The results displayed rate constants of 5.4 × 10^−5^ s^−1^ and 2.2 × 10^−4^ s^−1^, operating with Pt and BDD anode, respectively. For the record, some researchers have reported that the EO of organics might also follow a mixed first- and zero-order kinetics, in which the oxidation process is affected by both current and mass transfer limitations [93,94].

## 4. Cost Analysis and Energy Consumption 

In EO of organic contaminants, the removal efficiency of pollutants is usually estimated to evaluate the process viability. However, a meaningful subject in an EO process is the total operational cost required to treat the contaminants [101]. Total operating costs are usually related to electrical energy consumed, electrode supply and replacement, labor wages, pumping, stirring, cleaning, and maintenance procedures. As is well known, EO is considered an energy-consuming process; thus, the electric energy contribution is the more significant parameter to consider for estimating total operational costs. Moreover, electrode supply is a considerable capital cost driver in an EO process [12].

Electric energy requirement may be calculated in two different ways, depending on the organic concentration in the solution. The electric energy per mass (E_EM_) is used when the concentration of pollutants in the bulk phase is high [102]. Under this condition, the rate of degradation of organics can be considered a zero-order, because the removal of contaminants is directly dependent on the rate of electric energy applied. E_EM_ represents the electric energy in kilowatt-hours [kWh] essential to achieving the degradation of a unit mass of a contaminant in polluted water. If the EO process is performed in batch mode, E_EM_ is estimated by the following equation [102]:(17)$$E_{\mathrm{EM}}=\frac{Pt1000}{VM(C_{i}-C_{f})}$$
where *P* is the applied power (kW), *V* is the volume (L) of water treated, *t* is the time (h), *M* is the molar mass (g/mol) of the contaminant, and *c_i_* and *c_f_* are the contaminant concentrations at initial and final treatment time (mg/L), respectively.

If the EO process is performed in flow-through mode, E_EM_ is calculated through the following equation [102]:(18)$$E_{\mathrm{EM}}=\frac{Pt1000}{FM(C_{i}-C_{f})}$$
where *F* is the water flow rate (m^3^ h^−1^).

When the concentration of contaminant in the solution is low, it is recommended to use the electric energy per order (E_EO_). In this case, the degradation rate of pollutants can be considered a first-order, because the removal of contaminants depends on the rate of the electric energy provided to the system [102]. E_EO_ is the electric energy per time (kWh) required to remove a contaminant by one order of magnitude in a unit of volume (m^3^) of impacted water [12,102]. In batch mode, the following equation can be used [102]:(19)$$E_{\mathrm{EO}}=\frac{Pt1000}{Vlg(C_{i}-C_{f})}$$
where *lg*(*C_i_* − *C_f_*) is the logarithm of the difference of the concentrations.

In flow-through mode [102]:(20)$$E_{\mathrm{EO}}=\frac{P}{Flg(C_{i}-C_{f})}$$

The electrode cost contribution to the total operational costs in the EO process is usually estimated by the so-called anode surface area (ASA_O_) of the contaminant removed from the impacted water. ASA_O_ is calculated by the following equation [103]: (21)$${\mathrm{ASA}}_{O}=\frac{At}{Vlg(C_{i}-C_{f})}$$
where A is the apparent anode area (m^2^).

Table 5 lists operational cost assessments carried out by several authors. The total operating cost of the EO process mainly entails the cost of electrical energy and the cost of electrode required to treat 1 m^3^ of wastewater by EO at optimum operative conditions. 

However, it should be noted that the total operating costs are highly dependent on the treatment method, type, and initial concentration of the contaminant. Yet, the operation costs decrease by increasing the scale of the EO process acting in continuous configurations with respect to batch configurations [104].

## 5. Combination of EO with Photocatalysis

Although EO technology alone is effective, combining several AOPs techniques can further improve the removal of contaminants from wastewater and save energy [111,112]. For example, EO treatment of large volumes and amounts of water requires high energy consumption; hence, using technologies that need less energy is desirable. Numerous studies have been performed on combining EO with other AOPs techniques, which are called electrochemical advanced oxidation processes (EAOPs). Recently, several EAOPs have been employed for the treatment of impacted wastewater, such as electrocoagulation (EC), electro-Fenton (EF), photoelectro-Fenton (PEF), solar photoelectro-Fenton (SPEF), anodic oxidation with electrogenerated H_2_O_2_ (AO-H_2_O_2_), electrochemically activated persulfate processes, and photoelectrocatalysis (PEC), among others [113,114,115]. Although these techniques have shown technical feasibility to degrade recalcitrant contaminants, each exhibits several drawbacks. EC involves a high sacrificial anode replacement rate and maintenance needs [116]. Fenton-based EAOPs need to operate in acidic conditions, also involving a final step to remove the catalyst from the treated solution up to the legal limits [113]. AO-H_2_O_2_ requires a high amount of H_2_O_2,_ which must be met to achieve reasonable degradation efficiencies. Moreover, during electrochemically activated persulfate processes, radical sulphate must be activated by energy-based or electro-transfer-based methods, which may significantly impact the overall costs of the treatment [117].

Over recent years, PEC processes have gained more attention for degrading contaminants from aqueous media [118]. PEC can be an excellent solution to avoid the restrictions of solid–liquid separation, recycling of catalysts, and high energy consumption. Photocatalysis can help EO in the production of **^•^**OH, thereby increasing the removal efficiency of the system [119]. PEC can overcome the recombination rate of photogenerated electron-hole pairs, which constitutes one of the main limitations of simple photocatalysis processes [120]. Moreover, the possible involvement of natural light during the treatment makes it an environmentally friendly technique.

### Photoelectrocatalysis (PEC)

PEC is an AOPs considered environmentally friendly and highly efficient for degrading organic and persistent pollutants from wastewater. PEC consists of applying an external potential to a semiconductor film as a photocatalyst placed over a conductive substrate to avoid the recombination of photogenerated electron-hole (e^−^/h^+^) pairs. Thereby, the process increases h^+^ and the formation of hydroxyl radicals [121,122]. Oxidative–reductive reactions occurring during photocatalysis are shown in the following equations. They are divided into four steps, including excitation (Equation (22)), recombination, e^−^ scavenging (Equations (23) and (24)), and oxidation of hydroxyls (Equation (25)). The mechanism of photocatalysis consists of the initial photoenergy (*hv*) that excites a single electron to an empty electron band or the empty conduction band, leaving behind a positive hole (h^+^) [123,124]: photocatalyst + *hv* → e^−^ + h^+^(22)
e^−^ + h^+^ → e^−^ _(CB)_ + heat(23)
O_2(ads)_ + e^−^ → O_2_**^•^**^−^(24)
OH^−^ + h^+^ → **^•^**OH (25)

In the above equations, species such as **^•^**OH and O_2_**^•^**^−^ can react with contaminants and, as shown in Equations (26) and (27), can be responsible for removing contaminants from wastewater [125]: (26)$${•}^{\mathrm{OH}}+\mathrm{Pollutant}\;\to H_{2}O\;+\;CO_{2}$$
(27)$${O_{2}}^{•-}+\mathrm{Pollutant}\;\to H_{2}O\;+\;CO_{2}$$

The photoelectrocatalytic process combines electrochemical and photocatalytic processes. This means that electrochemical degradation and heterogeneous photocatalytic degradation are used simultaneously for the degradation of organic pollutants in wastewater, increasing the removal efficiency of pollutants due to the higher production of different radicals [126]. The photocatalyst mechanism is based on light energy (UV and solar/visible light). UV irradiation is more common than solar and visible light due to the large bandgap of the semiconductors [127]. When light (UV and solar/visible) is emitted to semiconductor catalysts and shines, it excites and activates them, eventually forming energy-rich electron-hole pairs. However, semiconductor catalysts should have suitable properties to increase efficiencies, such as the ability to absorb much light, low cost and high availability, and very good stability. To increase the efficiency of pollutant removal in the PEC process, operative conditions for both electrochemical and photocatalytic sections, such as pH, supporting electrolyte and concentration, light source, light intensity, photoelectrode type, and the thickness of semiconductor film, and the design of the photoelectrochemical reactor have to be optimized [128]. PEC is a reliable and well-known technique widely used since it has advantages such as low cost, high stability, green nature, non-toxicity, effective, and products with low pollutant load [129]. Table 6 shows a collection of recent articles focused on the PEC application for removing contaminants from wastewater.

## 6. Conclusions and Future Perspectives

This paper provided an overview of the EO process employed as an innovative method for removing emerging pollutants from water and wastewater. By reviewing a number of articles from recent years, the crucial role of several operational parameters has been critically examined. One of the advantages of further understanding the impact of these operational parameters on EO processes is reducing energy consumption and increasing the removal efficiency of pollutants. Finally, another important AOPs technique, called photoelectrocatalysis (PEC), widely used for degrading emerging pollutants, has been reviewed. 

This critical review revealed that the EO of emerging pollutants is rapidly progressing along its evolutionary path. However, there are still many deficiencies around it that need to be solved. A detailed study of the operating conditions is very critical, because a better understanding of the operating conditions will help to increase the efficiency of the process. The choice of appropriate anode material, having a high life cycle and efficiency, will play a significant role in the degradation of organic pollutants. Again, it is essential to pay attention to other operating parameters of electrochemical systems, such as the applied current intensity, the type and concentration of supporting electrolytes, the distance between the electrodes, the temperature, etc. Therefore, finding the best optimal points of operating conditions will be one of the main challenges of chemical EO of emerging pollutants. As a result of this overview, it is worth noting that one of the most important factors affecting the EO of chemicals is the configuration and the scale of the reactor selected. So far, many works have been performed with reactors on a laboratory scale, but it is very necessary to pay attention to the design of reactors on an industrial and large scale. Furthermore, electricity consumption and the cost of anode materials constitute the main factors affecting the economic feasibility of the EO process. Thus, identifying the best operating conditions can lead to a more cost-effective and efficient EO treatment.

Coupling both EO and photocatalysis allows for obtaining a combined PEC process capable of better treating pollutants in a more environmentally friendly way. However, the scarcity of such investigations over the recent years about PEC in real wastewater enhanced the need to deepen the application of this technology. Based on the results of this article, the future investigations of the researchers should be devoted to identifying the best operative condition, in terms of reactor design, for the process scaling-up, so shifting from lab-scale treatments to industrial processes capable of providing decentralized water treatment of urban and domestic wastewater. The next step of this technology should be to treat real wastewater containing several emerging pollutants at very low concentrations, thereby simulating actual environmental conditions. In the near future, the implementation of renewable energies will be pivotal to performing more sustainable PEC processes.

## Data Availability

No new data were created or analyzed in this study. Data sharing is not applicable to this article.

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
