# Peer review of "Application of Electrochemical Oxidation for Water and Wastewater Treatment: An Overview"

_molecules, 2023, doi:10.3390/molecules28104208_

Round 1

Reviewer 1 Report

It is important to consider some of the following points:

-In direct oxidation, the transformation reaction from R to CO2 is a bit ambitious, as this depends very much on e.g. if the anode is active or if the applied potential or current is high. Commonly, this is the general equation used in indirect oxidation, for example, produced on BDD electrodes where the production of hydroxyl radicals predominates as a consequence of the electrochemical oxidation of water, since due to its high oxidizing power, the organic matter is able to be completely mineralized to CO2. On the other hand, in non-active anodes, although the complete transformation to CO2 can occur, due to the low overpotentials of the oxygen evolution reaction, the transformation usually goes from R to RO.

-It is important to point out why during the review, emphasis is placed on the combination of EO with photocatalysis. Why other combinations such as EO with processes like Electro-Coagulation or process based on Fenton reaction like Electro-Fenton, Photo electro-Fenton, Solar photo electro-Fenton, in which very promising results have been obtained, are discarded? It is necessary to make clear the advantages of EO-PC over the other processes, in order to make it relevant in this literature review.

Author Response

The Authors wish to thank the Editor for the opportunity to improve and revise the manuscript entitled “Application of Electrochemical Oxidation for Water and Wastewater Treatment: an Overview” submitted to the Special Issue “Chemical Technologies for Environmental Analysis and Pollution Removal” on Molecules. The Authors considered all the Reviewers’ comments and suggestions, and significant corrections have been made to address each point and improve the article. This cover letter has been prepared to explain point-by-point the details of the revisions in the manuscript and the Authors’ responses to the Reviewers’ comments. The detailed response to the Reviewers’ comments, not including typo corrections, is reported below.

Reviewer 1

The Authors wish to thank the Reviewer for his/her constructive comments, helpful words and the time spent revising the manuscript. All the Reviewer’s suggestions were considered, and some corrections were made to improve the manuscript.

1) Reviewer’s comments and suggestions for Authors

In direct oxidation, the transformation reaction from R to CO2 is a bit ambitious, as this depends very much on e.g. if the anode is active or if the applied potential or current is high. Commonly, this is the general equation used in indirect oxidation, for example, produced on BDD electrodes where the production of hydroxyl radicals predominates as a consequence of the electrochemical oxidation of water since due to its high oxidizing power, the organic matter is able to be completely mineralized to CO2. On the other hand, in non-active anodes, although the complete transformation to CO2 can occur, due to the low overpotentials of the oxygen evolution reaction, the transformation usually goes from R to RO.

Authors’ response

The Authors agree with the comment and thank the Reviewer for pointing this out. The direct EO mechanism reported in section 3.1 Direct EO of the main manuscript has been rediscussed based on the above suggestions. The changes are tracked into the revised version of the manuscript using MS Word Tracking Mode.

3.1 Direct EO (original version)

Direct EO is based on the so-called electrochemical oxygen transfer reaction (OTR), where oxygen is transferred from water to the organic molecules by means of electric energy:

R + H2O → CO2 + H+ + e-                                                                                                                 (3)

R is a general organic compound.

WAS CHANGED IN

3.1 Direct EO (revised version)

The direct EO mechanism only involves the contribution of electrons, which oxidize the contaminants molecules, resulting in direct charge transfer reactions[1]. The first step is the adsorption of the pollutants onto the anode surface. The process is mainly controlled by the molecule transport and electron transfer rate at the anode/solution interface2:

R + M → M-Rads                              (adsorption reaction)                    (3.a)

M-Rads + e- → M-Rads,ox         (direct charge electron reaction)                 (3.b)

M-Rads,ox → Rox                                (desorption reaction)                                       (3.c)

where R is the pollutant molecule, and M is referred to the anode surface.

[1] He, Y.; Lin, H.; Guo, Z.; Zhang, W.; Li, H.; Huang, W. Recent Developments and Advances in Boron-Doped Diamond Electrodes for Electrochemical Oxidation of Organic Pollutants. Sep. Purif. Technol. 2019, 212, 802–821, doi:10.1016/j.seppur.2018.11.056.

2 Garcia-Segura, S.; Ocon, J.D.; Chong, M.N. Electrochemical Oxidation Remediation of Real Wastewater Effluents — A Review. Process Saf. Environ. Prot. 2018, 113, 48–67, doi:10.1016/j.psep.2017.09.014.

Reviewer 2 Report

The authors propose a review of a vast area of ​​research that has been developing intensely. Due to this characteristic, it is important that 

1) the authors explain the parameters used in the bibliographic research;

2) Normalize all units: current density in mA cm-2 (transform those with different units);

3) For electric current, the capital letters I and J are used, while for electric current density, the corresponding lowercase letters are used;

4) The combination of methods has been an aspect of growing interest, but only photoelectrocatalysis has been addressed. Comment a little more on method combinations;

5) I suggest the analysis of the following recent manuscript: https://doi.org/10.1002/appl.202300008

Author Response

The Authors wish to thank the Reviewer for his/her constructive comments and for the time spent revising the manuscript. All the Reviewer’s suggestions were considered, and some corrections were made to improve the manuscript.

1) Reviewer’s comments and suggestions for Authors

The authors explain the parameters used in the bibliographic research.

Authors’ response

The Authors would like to thank the Reviewer for his helpful remark. An explanation of the applied search strategy has been added to the Section 1.1. Methodology and Search strategy. The changes are tracked into the revised version of the manuscript using MS Word Tracking Mode.

2) Reviewer’s comments and suggestions for Authors

Normalize all units: current density in mA cm-2 (transform those with different units).

Authors’ response

The Authors would like to thank the Reviewer for his recommendation. Over the entire manuscript, the current density unit has been expressed as mA/cm2. The modifications are tracked into the manuscript by using MS Word Tracking Mode.

3) Reviewer’s comments and suggestions for Authors

For electric current, the capital letters I and J are used, while for electric current density, the corresponding lowercase letters are used.

Authors’ response

The authors thank the Reviewer for the suggestions. Lowercase letter i and the capital letter I were used over the whole manuscript for electric current density and electric current, respectively. The modifications are tracked into the manuscript by using MS Word Tracking Mode.

4) Reviewer’s comments and suggestions for Authors

The combination of methods has been an aspect of growing interest, but only photoelectrocatalysis has been addressed. Comment a little more on method combinations.

Authors’ response

The Authors wish to thank the Reviewer for the interesting suggestions. Combined electrochemical advanced oxidation processes (EAOPs) were discussed in Section 5. Combination of EO with photocatalysis. Moreover, Sections 1. Introduction and 6. Conclusions and future perspectives were also briefly modified to highlight better the choice of the Authors to explore EO-PC combined process. The changes are tracked into the revised version of the manuscript using MS Word Tracking Mode.

5) Reviewer’s comments and suggestions for Authors

I suggest the analysis of the following recent manuscript: https://doi.org/10.1002/appl.202300008.

Authors’ response

The Authors would like to thank the Reviewer for his/her helpful remark and the useful reference suggested, which was included in Sections 1. Introduction and 2.1.2. Effect of the supporting electrolyte and electrolyte concentration of the manuscript. The changes are tracked into the revised version of the manuscript using MS Word Tracking Mode.

Round 2

Reviewer 1 Report

The manuscript with the actual corrections in relation with the reactions 3a-3c it is ok for me. I appreciate the considerations in this.